# Coat, Claw and Dewclaw 17-β-Estradiol and Testosterone Concentrations in Male and Female Postpubertal Cats: Preliminary Results

**DOI:** 10.3390/ani13030522

**Published:** 2023-02-01

**Authors:** Jasmine Fusi, Tanja Peric, Monica Probo, Roberta Bucci, Massimo Faustini, Maria Cristina Veronesi

**Affiliations:** 1Department of Veterinary Medicine and Animal Sciences, Università degli Studi di Milano, 26900 Lodi, Italy; 2Department of Agricultural, Food, Environmental and Animal Sciences, University of Udine, 33100 Udine, Italy; 3Faculty of Veterinary Medicine, Veterinary Teaching Hospital University of Teramo, 64100 Teramo, Italy

**Keywords:** coat, claws, dewclaws, 17-β-estradiol, testosterone, postpubertal cats

## Abstract

**Simple Summary:**

In the study of long-term hormonal changes, matrices such as hair/coat and nails/claws proved to be useful and collectable without invasiveness, respecting animal welfare. This study aims to verify the usefulness of coat, claws and dewclaws to measure the main sexual steroids 17-β-estradiol (E2) and testosterone (T) concentrations in male and female postpubertal domestic cats during the breeding season. A total of 20 male and 18 female postpubertal domestic cats were enrolled and sexual steroids measured in coat and claws collected only once during the neutering/spaying procedures. Coat was collected from the forearm (ACOAT) and from the surgical area (SCOAT). Dewclaws (DCLAWS) were collected separately from the claws of all the other digits of the forearms (CLAWS). All the tissue materials were suitable for measuring both sexual steroids, with both ACOAT and SCOAT useful for distinguishing between males and females on the basis of T concentrations. Within each sex, E2 and T concentrations can be assessed on coat, but also on the dewclaws, providing an alternative, practical, matrix for sexual steroid measurement in postpubertal cats during the breeding season.

**Abstract:**

In the recent past, tissue materials such as hair/coat and nails/claws have proved to be useful for the study of long-term hormonal changes in humans and animals and shown to be advantageous in terms of being collectable without invasiveness, with a benefit in terms of animal welfare. However, studies using these tissue materials in cats are scarce, especially on sexual hormone measurement. In this study, the concentrations of 17-β-estradiol (E2) and testosterone (T) were assessed in 20 male and 18 female domestic postpubertal cats at the time of neutering/spaying during the breeding season. Hormones were measured in coat shaved from the forearm (ACOAT) and from the surgical area (SCOAT); claws were collected from the forearms (CLAWS) and the dewclaws (DCLAWS). Although all these tissue materials were shown to be useful for E2 and T long-term measurement, only T concentrations were higher (*p* < 0.001) in males from both ACOAT and SCOAT samples when compared to females and therefore useful for distinguishing between the two sexes. Within each sex, E2 and T concentrations can be assessed on coat, but also on the dewclaws, providing an alternative, practical, matrix for sexual steroid measurement in postpubertal cats during the breeding season.

## 1. Introduction

In domestic animals, all the reproductive functions are under the major control of sexual steroids, secreted by the gonads [1]. In both males and females, sexual steroid secretion is controlled by the hypothalamic–pituitary–gonadal axis, developed before birth, briefly active immediately after birth and then quiescent until puberty [2].

Sexual steroid hormones are traditionally measured with blood tests, a first-line choice in many aspects of small animal reproduction. However, for monitoring long-term hormonal production, the necessity of daily repeated exams makes this way of sampling less convenient. Similarly, other matrices such as saliva, urines and feces, collectable without invasiveness, also provide only a momentary or short-term picture of measured hormones. Moreover, all these matrices need to be stored by freezing when not analyzed immediately after collection [3,4]. In this context, when long-term steroid hormone changes have to be investigated, hair/coat and nails/claws represent the first choice of tissue materials. Many studies have in fact proved the reliability of hair/coat and nails/claws for the retrospective evaluation of long-term hormonal accumulation in both humans and animals [5,6,7,8,9], including felids [10,11,12,13,14,15], especially for cortisol and dehydroepiandrosterone (DHEA). Other than their reliability, the non-invasive collection of these matrices has the advantage of respecting animal welfare [9,16]. Moreover, these tissue materials are stable at room temperature and storable without freezing process [17]. Regarding hair/coat, although some studies have debated the influence of different areas of collection on coat cortisol concentrations in different animal species [10,18,19,20], including cats [13], almost no information is available regarding sexual steroids. In a study on coyote puppies, [21] Schell and colleagues demonstrated the absence of a difference in testosterone or cortisol concentrations in coat collected from different sampling areas.

In addition to hair/coat, Palmeri and colleagues [22] demonstrated the mechanism by which compounds are incorporated in the nails, and [23] showed, in humans, the positive correlation between lipophilicity of the compound and its affinity with the keratin of nails, which enhances the accumulation of lipophilic hormones into this matrix. Therefore, sexual steroids such as estrogens and testosterone, reported to stably bind to keratin, could be usefully measured in nails [23]. Although one could suppose that nails/claws are externally very different among species, they were instead reported to be very similar from an anatomic point of view, enclosing in all cases the distal phalanx [24]. However, differently from many other species, the peculiarity of cat claws is that they can be retracted or extracted voluntarily by the subjects [25]; this means that they are protected from wear and could grow quicker than human nails [26]. Another characteristic of cats’ claws is the claw shedding [25], a phenomenon by which, periodically, they lose the external cornified claw sheath, at the same time maintaining the internal layers intact [25]. However, the forelimbs of cats usually have a first pair of claws, called dewclaws, which are not retractable and are not involved in weight bearing during walking, but which are reported to be used for pray catching [27].

Despite this reasonable advantage of this species, to date, hormonal investigations using these tissue materials in domestic cats are still scanty. A recent study [13] showed that hair collected from the lumbosacral area and the nails could be used for the assessment of cortisol concentrations for measuring stress in cats. The same authors also reported a growth rate in the front nails of cats of 2.4 mm in 21 days, even if the portion of trimmed nails may contain the hormones accumulated in a prior time period (at least 1 month) [13]. Hair was also used for cortisol measurement in cats in other studies [5,15], and in feral and wild felids [11]. Very recently, [14] investigated the usefulness of seasonal coat collection in wild felids and domestic cats for the measurement of not only cortisol, but also testosterone concentrations. The authors found annual dynamic fluctuations, influenced by the studied species, and also interesting sex-related differences in coat testosterone concentrations during the breeding season in domestic cats.

A picture of the long-term accumulation of sexual steroid hormones, measurable in a single collection of coat and claws, could be a supplementary tool for the diagnosis confirmation of some reproductive disturbances, such as delayed puberty, estrus or male sexual behavior in gonadectomized cats and unilateral gonadectomized cryptorchid cats, when traditional investigations provide unclear results. However, before a real and practical application, specific physiologic values for both tissue materials in different reproductive status and in both sexes are needed.

Therefore, because of all the above-reported reasons, the aims of the present study were to assess (1) the usefulness of a single collection of two different sampling sites of coat, of claws and of dewclaws for the measurement of long-term accumulation of E2 and T in postpubertal domestic cats during the breeding season; (2) possible differences among the two coat sampling sites, claws and dewclaws in the concentrations of E2 and T between postpubertal male and female cats during the breeding season; and (3) possible differences in the concentrations of E2 and T within each sex among the two coat sampling sites, claws and dewclaws in postpubertal male and female cats during the breeding season.

## 2. Materials and Methods

### 2.1. Ethics

The study was approved by the Università degli Studi di Milano Ethical Committee (OPBA) with protocol OPBA_117_2022. All the owners signed a written informed consent to the collection of coat, claws and dewclaws for research purposes.

### 2.2. Animals

A total of 20 male (median age 1 year; min max: 9 months–10 years) and 18 female (median age 1 year; min max: 8 months–10 years) adult domestic, indoor-housed cats (Felis catus) belonging to private owners were enrolled on the day of neutering/spaying, between June and August 2022, at the Veterinary Teaching Hospital of Università degli Studi di Milano—Faculty of Veterinary Medicine. They were of different feline breeds, healthy and in good general condition. From a reproductive standpoint, all the subjects were normal, without genital diseases/malformations, and they had never been treated with medications or hormones that could have interfered with their reproductive status.

Inclusion criteria for postpubertal ascertainment were: age of the cats (>8 months); evidence of puberty such as the occurrence of the first heat in females, observed at least two months before entering the study, whilst in males the observation of sexual behavior, with urine spraying, at least two months before entering the study, with evidence of penile spines presence at the time of study.

### 2.3. Coat and Claws Collection

During the surgical preparation of the patient, each cat was submitted to coat and claw collection. To standardize the method and the results of coat and claw collection, in all cases, coat and claws were collected by the same author (JF).

Coat was collected from two different sampling sites: by shaving an area of about 3 cm^2^ from the dorsal surface of the forearm (ACOAT) and from the surgical area—scrotum in the males and ventral abdomen in females (SCOAT). Shave was performed by a razor (TN2300 Nomad, Rowenta spa, Milan, Italy) to allow the collection of the coat at the level of the skin. The two collected coat samples were immediately placed into separated individually coded paper envelopes and stored in the dark at room temperature until analysis.

The tips of the claws belonging to the 1st digit of both forelegs were collected by clipping, and then, the two dewclaws were immediately placed in a single individually coded paper envelope and stored in the dark at room temperature until analysis (DCLAWS). The tips of all the other claws of both forelegs were collected and stored in a single individually coded paper envelope (CLAWS) as descripted above for dewclaws. After every coat, claw and dewclaw collection, the razor and the claw clipper were disinfected with a 70% alcohol solution [28].

### 2.4. Hormonal Analysis

Coat strands, claws and dewclaws were washed in 3 mL isopropanol to ensure the removal of any steroids from their surface. Coat (25 mg of sample), claw (15 mg of sample) and dewclaw (10 mg of sample) steroids were than extracted with methanol, as already described [8,29] for dogs, and the concentrations of 17-β-estradiol and testosterone were measured using a solid-phase microtiter RIA. In brief, a 96-well microtiter plate (OptiPlate; Perkin-Elmer Life Sciences, Boston, MA, USA) was coated with goat anti-rabbit γ-globulin serum diluted 1:1000 in 0.15 mM sodium acetate buffer (pH 9) and incubated overnight at 4 °C. The plate was then washed twice with an RIA buffer (pH 7.5) and incubated overnight at 4 °C with 200 μL of the antibody serum diluted 1:80,000 for 17-β-estradiol and 1:160,000 for testosterone. The rabbit 17β-estradiol and testosterone antibodies used were obtained from Analytical Antibodies (Bologna, Italy). The cross-reactivities of the anti-17-β-estradiol antibody with other steroids were as follows: 17β-estradiol, 100%; estrone, 2.5%; estriol, 0.12%; 17β-estradiol-(B—D-glucuronide), 0.04%; 17β-estradiol-3-sulfate 0.012%; DHEA, 0.007%; 17α-estradiol, <0.04%; progesterone, <0.04%; testosterone, <0.04%; androstenedione, <0.04%; estrone-3-sulfate, <0.04%. The cross-reactivities of the anti-testosterone antibody with other steroids were as follows: testosterone, 100%; 5α-dihydrotestosterone, 43.2%; 5α-androstanedione, 33.1%; 5β-androstanedione, 11.4%; 5α-androstan-3α,17β-diol, 9.4%; androstenedione, 0.4%; testosterone 17β-glucuronide, 0.09%; progesterone, DHEA, 17β-estradiol androsterone-3-glucuronide, 0.01%; androsterone-3-glucuronide, 0.006%; cortisol, <0.001%. After washing the plate with RIA buffer, the standards (5–200 pg/well), the quality-control extract, the test extracts and the tracer (17-β-estradiol [2,4,6,7,16,17-3H (N)]; Amersham/Searle/GE Healthcare Life Sciences (UK) or testosterone [1,2,6,7-3H (N)]; Perkin-Elmer Life Sciences, Boston, MA, USA) were added, and the plate was incubated overnight at 4 °C. The bound hormone was separated from the free hormone by decanting and washing the wells in RIA buffer. After the addition of 200 μL of scintillation cocktail, the plate was counted on a β-counter (Top-Count; PerkinElmer Life Sciences, Boston, MA, USA). The intra- and inter-assay coefficients of variation were 3.7 and 12.1 for 17-β-estradiol and 4.40 and 11.5% for testosterone, respectively. The sensitivities of the assays were 15.40 pg/mL and 6.60 pg/mL for 17-β-estradiol and testosterone, respectively.

### 2.5. Statistical Analysis

After data normality distribution assessment through the Shapiro–Wilk test, the non-parametric Kruskal–Wallis test was used to verify possible differences between E2 and T concentrations in ACOAT, SCOAT, DCLAWS and CLAWS between male and female postpubertal cats. Subsequently, the Dwass–Steel–Critchlow–Fligner test for pairwise comparisons was used to assess possible differences in E2 and T concentrations among the ACOAT, SCOAT, DCLAWS and CLAWS within each sex (males and females). Statistical significance was set at *p* < 0.05 (Jamovi project 2022, ver 2.3.13, retrieved from https://www.jamovi.org accessed on 3 October 2022).

## 3. Results

The collection of the coat, claws and dewclaws was always successfully and easily performed. The mean ± SD individual amounts of coat samples were 90.6 ± 24.4 mg (range: 17.3–126.8 mg); those of claw samples were 48.9 ± 27.1 mg (range: 9.6–122.9 mg); and those of dewclaw samples were 8.6 ± 4.5 mg (range: 1.9–24.4 mg); all tissue materials in all cases allowed the analysis of both E2 and T.

Data, expressed as median and min–max values, regarding E2 and T concentrations in ACOAT, SCOAT, DCLAWS and CLAWS in male and female postpubertal cats are reported in Table 1.

The statistical analysis showed significant higher T concentrations in SCOAT and ACOAT in males in comparison to females. Within each sex, a higher concentration of E2 and T in SCOAT, ACOAT and DCLAWS than in CLAWS was detected, but without significant differences between SCOAT, ACOAT and DCLAWS.

## 4. Discussion

To the authors’ knowledge, this study reports, besides the confirmed usefulness of coat and claws, the first use of dewclaws as a possible matrix for the measurement of E2 and T in postpubertal domestic cats during the breeding season. The results, in fact, demonstrated once more the usefulness of coat for studies monitoring function for an extended period, as previously suggested in humans and animals, and specifically in felids [5,11,14,15], even if more studies are needed to discover the exact timing and percentage of hormone accumulation from peripheral blood. Besides coat, the present study also confirmed that claws could be suitable for measuring sexual steroids in postpubertal cats, supporting the first reports about the usefulness of claws for non-invasive and objective hormonal assessment in cats [12,13]. However, the present study preliminarily highlighted, for the first time, the usefulness of dewclaws alone for assessing E2 and T concentrations in postpubertal cats, even if it did not allow for distinguishing between males and females.

### 4.1. Coat and Claws for Differentiation between Male and Female Postpubertal Cats

Although specific reference values could not be drawn from these results, given the small number of subject enrolled, some preliminary data were obtained. In fact, the present study results showed that coat T concentration measurement is suitable for distinguishing between male and female postpubertal cats during the breeding season, as a possible future tool additional to clinical or blood hormone investigation. Moreover, the analysis performed on matrices characterized by long-term accumulation could probably overcome the issue related to the high T fluctuations measured in blood [30]. This result agrees with the recently reported seasonal increase in T concentrations in the coat of male domestic cats [14] and demonstrates that a difference between male and female cats can be observed. However, the opposite, that is, a higher E2 coat concentration in females, was not detected, although all the females were sampled during the breeding season when high concentrations of circulating E2 can be expected. In the present study, however, differently to the findings reported by [13] on cortisol accumulation, no significant differences in E2 and T concentrations were found between the two coat sampling areas (forearm or scrotum/abdomen). This result highlights that a similar accumulation of sexual hormones seems to occur in these two different coat areas of the body and suggests that the forearm should be considered a first choice for the sampling of coat due to the easier way of collection compared to the scrotal/abdominal area, especially when samples from enrolled animals are collected while animals are awake. Claws and dewclaws were not shown to be useful for distinguishing between sexes on the basis of T concentrations. Besides T, the measurement of E2 concentrations in these matrices was not useful for the distinction of sexes. In fact, similarly to results obtained in coat, almost superimposable E2 concentrations were detected in males and females within each matrix. This highlights the need for further investigations with the enrollment of a higher number of subjects, taking into account that cats are seasonal breeders.

### 4.2. Coat, Claws and Dewclaws as Matrices for the Assessment of E2 and T Concentrations within Female and Male Postpubertal Cats

When female and male postpubertal cats were considered separately, coat and dewclaws were shown to be more useful than claws for the assessment of E2 and T concentrations. Therefore, as mentioned above, the forearm area should be considered the first choice of sampling for coat collection. However, it is interesting to note that both dewclaws and coat are useful for E2 and T concentration assessment, even if dewclaws in the present study fail in distinguishing the sex. This finding is also important when collection of samples is considered, with the collection of dewclaw samples being faster and easier than that of forearm coat samples. The use of declaws could help to increase the number of subjects enrolled, as its collection could even be performed during clinical examination, and not only when coat is shaved for procedures such as blood samplings or insertion of a venous catheter. A final comment regards the lowest usefulness of the claws for E2 and T concentration assessment in both sexes. From a practical standpoint, this preliminary finding is interesting because the collection of claws from all digits of the forearms could be less practical than the collection of only dewclaws in animals that are awake, considering the different time of handling needed, not always tolerated by cats. At a first glance, this could be considered in disagreement with a study performed in humans, in which no significant differences in T concentrations in nails collected from the right and left hand were found [23]. However, although claws and dewclaws could be considered very similar in their structure, dewclaws have the advantage of being less exposed to scratching consumption and damage than other claws and could therefore better reflect the previous hormone accumulation. Finally, while claws of the forearm have been reported to have a growth rate of 2.4 mm within 21 days [13], to the authors’ knowledge, no specific data are available regarding the growth rate of dewclaws. In this case, further studies are also needed to define the specific growth rate of dewclaws, additionally evaluating whether the type of housing (indoor or outdoor) can influence the growth rate of dewclaws.

## 5. Conclusions

In conclusion, the results of the present preliminary study demonstrated that a single coat, claw and dewclaw collection can be useful for the measurement of E2 and T in postpubertal cats during the breeding season. However, the different accumulation of the sexual steroids studied in the diverse tissue materials is interesting and deserves further investigations. When aiming to distinguish between postpubertal males and females, the measurement of T concentrations in coat, independently from the sampling site, proved to be the only useful matrix. Therefore, due to the easiest collection of coat from the forearm in comparison to the scrotal/abdominal area, the former should be considered the first-choice sampling site, especially in patients that are awake. Within each sex, E2 and T concentrations can be assessed on coat, but also on dewclaws. This study opens new frontiers for the use of innovative matrices; however, future studies with the enrollment of more subjects, especially during the breeding season, will help to define the concentration ranges necessary for the practical use of coat and claws.

## Figures and Tables

**Table 1 animals-13-00522-t001:** Data regarding E2 and T concentrations (pg/mg) (median and min–max values) in SCOAT, ACOAT, DCLAWS and CLAWS in male and female postpubertal cats.

	SCOAT		ACOAT		CLAWS		DCLAWS	
SEX	E2 (pg/mg)	T (pg/mg)	E2 (pg/mg)	T (pg/mg)	E2 (pg/mg)	T (pg/mg)	E2 (pg/mg)	T (pg/mg)
Males	1.1 ^A^ (0.8–1.6)	2.0 ^aC^ (1.1–4.2)	1.2 ^A^ (0.7–1.6)	1.7 ^aC^ (1.0–3.0)	0.4 ^B^ (0.2–1.1)	0.4 ^D^ (0.2–1.1)	1.3 ^A^ (0.5–3.8)	1.1 ^C^ (0.5–7.5)
Females	1.1 ^A^ (0.9–1.6)	1.0 ^bC^ (0.6–2.0)	1.1 ^A^ (1.0–1.8)	1.0 ^bC^ (0.7–3.4)	0.5 ^B^ (0.04–1.5)	0.4 ^D^ (0.2–1.7)	1.3 ^A^ (0.3–6.4)	1.1 ^C^ (0.4–3.6)

^a^,^b^ differences within column *p* < 0.001; ^A^,^B^ differences in E2 within rows *p* < 0.001; ^C^,^D^ differences in T within rows *p* < 0.001.

## Data Availability

Data are available upon reasonable request to the corresponding author.

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
