# Peer review of "Coat, Claw and Dewclaw 17-β-Estradiol and Testosterone Concentrations in Male and Female Postpubertal Cats: Preliminary Results"

_animals, 2023, doi:10.3390/ani13030522_

Round 1

Reviewer 1 Report

It is interesting and novel study, although quite preliminary considering the number of animals used. I have only a few minor concerns to be addressed before considering for publication:

 I would suggest to add a word pilot/preliminary in the title

Please simplify (mainly language and the depth of description) in the simple summary part

Please change:

measured on blood for measured in blood

episodic manner in pulsatile manner

Please consider changing ‘matrices’ into bodily fluids or tissue/material/source where appropriate

Line 84-85 please add the names of the cited authors

Line 92: please re-write/change ‘consumption’

Line 286-289:  please consider that ‘practical point of view’ and ‘routine visit’ are a bit of an overstatement, the collection is quite easy, but the same cannot be said about the lab work necessary the hormonal evaluation

Author Response

REVIEWER 1

Comments and Suggestions for Authors (indicated as "AU" below)

It is interesting and novel study, although quite preliminary considering the number of animals used. I have only a few minor concerns to be addressed before considering for publication:

I would suggest to add a word pilot/preliminary in the title
AU: The words “preliminary results” were added as suggested
Please simplify (mainly language and the depth of description) in the simple summary part
AU: The simple summary was simplified as suggested

Please change:
measured on blood for measured in blood
AU: changed as suggested

episodic manner in pulsatile manner
AU: changed as suggested

Please consider changing ‘matrices’ into bodily fluids or tissue/material/source where appropriate AU: “Matrices” was changed all over the text where appropriate

Line 84-85 please add the names of the cited authors
AU: Added as suggested

Line 92: please re-write/change ‘consumption’
AU: changed as suggested

Line 286-289: please consider that ‘practical point of view’ and ‘routine visit’ are a bit of an overstatement, the collection is quite easy, but the same cannot be said about the lab work necessary the hormonal evaluation

AU: this part was modified according to the reviewer’s suggestion

Reviewer 2 Report

The present study examined the usefulness of haircoat, claws, and dewclaws as alternatives to blood for measuring sex steroids in pubertal domestic cats. The investigators report that haircoat and dewclaws are good sources for measuring sex steroids, but claws have little usefulness. None of the sources allowed the separation of male and female by testosterone concentration. In contrast, none of the sources were useful for separating males and females by estrogen concentrations. Of note, assay sensitivities are reported in pg/ml, and data are reported in pg/mg tissue. Limitations of the study are that samples were only from pubertal animals and only during the breeding season and not compared to blood. Haircoats had greater testosterone concentrations than dewclaws, but estrogen concentrations were not different.  The investigators overstated the utility of dewclaws. 

Author Response

The authors are grateful to the Reviewers for criticisms and suggestions that ameliorate the quality of the manuscript.

All the suggestions/changes have been accepted and text changed accordingly.

In attachment the point-by-point answers to each Reviewer.

REVEWER 2

Comments and Suggestions for Authors (below indicated as "AU")

The present study examined the usefulness of haircoat, claws, and dewclaws as alternatives to blood for measuring sex steroids in pubertal domestic cats. The investigators report that haircoat and dewclaws are good sources for measuring sex steroids, but claws have little usefulness.

None of the sources allowed the separation of male and female by testosterone concentration.

AU: The study showed that only coat allowed to distinguish between male and female cats by measuring testosterone concentrations.

In contrast, none of the sources were useful for separating males and females by estrogen concentrations. Of note, assay sensitivities are reported in pg/ml, and data are reported in pg/mg tissue.

AU: The sensitivity of the assay is given by the standard curve that is always expressed in mL, given that the hormonal concentration in the standard is defined in relation to mL. Nevertheless, the samples we tested for hormonal concentrations are solid matrixes and thus it is correct to express the hormone concentration per mg of sample analysed.

Limitations of the study are that samples were only from pubertal animals and only during the breeding season and not compared to blood.

AU: As one of the first study on this topic, we choose to enroll only pubertal and within breeding season cats in order to assess the hormonal concentrations in animals in which a current hormonal stimulation was supposed. We agree with the reviewer that it will be interesting to enlarge the study to other reproductive status and during the whole year.

About the comparison with the blood, the goal of this study was to evaluate alternative non-invasively collected samples. Thus, to not analyze blood was deliberately chosen in building our experimental plan. In the past many studies tried to correlate retrospective matrices (hair, nails, ...) to punctual matrices (blood); some of them found correlations but many failed in this intention. Hair is a retrospective matrix that incorporates hormone during its growth (anagen phase) when the follicular cells are strictly connected with the capillaries and thus with the systemic circulation of hormones. After that, a lag-time occurs, namely a period in which that section of the hair (or claw) needs to emerge (being available for the cutting) (indeed, our study is based on cut and not pulled samples). Blood would instead give us a punctual information regarding the instant in which we are sampling from the systemic circulation. Thus, comparison with blood would mean trying to compare two different kinds of information (different from the temporal point of view): one regarding the past (hair, claws, ...) and one regarding a flash (single point in time) got from blood. Therefore, we would not expect any correlation between them.

Haircoats had greater testosterone concentrations than dewclaws, but estrogen concentrations were not different. The investigators overstated the utility of dewclaws.

AU: The authors agree with the reviewer, and statements about dewclaws utility were rewritten more cautiously.

Reviewer 3 Report

The manuscript “Coat, claws and dewclaws 17-β-estradiol and testosterone concentrations in male and female pubertal cats” describes the use of alternative biomatrices such as haircoat and claws in cats for the measurement of 17-β-estradiol and testosterone. 

Major concerns:

The authors justify the use of these matrices for E2 and T measurement in female and male cats by saying that “A picture of the long-term accumulation of sexual steroid hormones, measurable in a single collection of coat and claws, could be a promising diagnostic tool for some reproductive disturbances, such as delayed puberty, estrus or male sexual behavior in gonadectomized cats, and unilateral gonadectomized cryptorchid cats. However, before a real and practical application, specific physiologic values for both matrices, in pubertal male and female cats are needed.” (Lines 110-115). However, in my opinion, the potential application of such matrices as clinical diagnostic tools for the aforementioned clinical conditions is questionable. There are other, more practical and accurate diagnostic approaches available for the work-up of those conditions, i.e. serum AMH, progesterone, vaginal cytology, ultrasound, presence or absence of penile spines, etc., which should precede or exclude the use of T or E2 measurement from haircoat or claw.

I disagree with the authors’ statement (Lines 66-71) that “Although sexual steroids were traditionally measured on blood, this matrix could be not considered as a first-line choice, because of invasiveness at collection and because of the only punctual information provided, not useful for monitoring hormones secreted in an episodic manner, like testosterone. Similarly, also other matrices like saliva, urines and feces, collectable without invasiveness, provide only punctual or short-term picture of measured hormones.” In many instances, we do need to measure steroid hormone levels from blood at a given point in time. While due to the episodic release of T or E2 this may not always be useful, even a baseline T could often differentiate intact from gonadectomized males, or a vaginal cytology can give accurate and useful information within minutes about the presence of estrogen influence in the body instead of blood E2 in cats. Therefore, I question the rationale and practicality of measuring sex steroids in cats in these alternative matrices, and whether such long-term accumulated levels could have a diagnostic application. Also, we do not really know for how long and at what concentrations these matrices are able to store information about the long-term levels of these steroid hormones, and how these levels are relevant to certain reproductive events and time points.

The authors’ results show that there are differences in hormone concentrations between the claw and dewclaw sites, but there are no differences between males and females in T levels in the claw/dewclaw matrices in contrast to the haircoat. This finding is in contrast to the authors’ conclusion that dewclaws are useful for E2 and T measurements (Lines 250-251), if it cannot even differentiate between males and females.

Line 255-257: “In fact, the present study results showed that coat T concentrations measurement is suitable to distinguish between male and female pubertal cats during the breeding season.” Again, while it is scientifically possible, why should we use such matrices and measure hormones for this simple task?

As for the selection of animals, the authors define their study population as “pubertal”, however, they have animals from 8-9 months to 10 years of age. By the authors’ description it seems that all animals were already postpubertal, as pubertal would rather refer to those animals that are going through puberty at that time. Also, the authors would need to specifically state whether these cats were reproductively normal, e.g. no cryptorchidism, DSD, ovarian/testicular pathology, etc, and whether or not they received (currently or in the past months) any medications /hormones that could interfere with reproductive status and hormone levels.

Minor points of concern:

Lines 46-61 of Introduction: This description of physiology can be deleted.

Line 64: HPG (gonadal) axis, not HPA.

Please revise the English, some sentences are difficult to understand.

Line 165-166: Was the amount of coat/claw used for the extraction the same across samples? Please describe.

Table 1: Please clarify if concentrations are expressed on the freeze-dried weight.

Table 2 and 3 could be deleted and the results denoted in Table 1 or explained within the text.

Author Response

The authors are grateful to the Reviewers for criticisms and suggestions that ameliorate the quality of the manuscript.

All the suggestions/changes have been accepted and text changed accordingly.

In attachment the point-by-point answers to each Reviewer.

REVIEWER 3

Comments and Suggestions for Authors (below indicated as "AU")

The manuscript “Coat, claws and dewclaws 17-β-estradiol and testosterone concentrations in male and female pubertal cats” describes the use of alternative biomatrices such as haircoat and claws in cats for the measurement of 17-β-estradiol and testosterone.

Major concerns:

The authors justify the use of these matrices for E2 and T measurement in female and male cats by saying that “A picture of the long-term accumulation of sexual steroid hormones, measurable in a single collection of coat and claws, could be a promising diagnostic tool for some reproductive disturbances, such as delayed puberty, estrus or male sexual behavior in gonadectomized cats, and unilateral gonadectomized cryptorchid cats. However, before a real and practical application, specific physiologic values for both matrices, in pubertal male and female cats are needed.” (Lines 110-115). However, in my opinion, the potential application of such matrices as clinical diagnostic tools for the aforementioned clinical conditions is questionable. There are other, more practical and accurate diagnostic approaches available for the work- up of those conditions, i.e. serum AMH, progesterone, vaginal cytology, ultrasound, presence or absence of penile spines, etc., which should precede or exclude the use of T or E2 measurement from haircoat or claw.

AU: The authors agree with the Reviewer about the fact that we were not clear enough and have overstated the usefulness of the present study results as a now-available diagnostic tool. We hope that this method could be, in the future, supplementary to the clinical or blood hormones investigations presently available, when necessary. In fact, the authors agree with the Reviewer that, at present, the clinician could use clinical investigations (such as vaginal cytology or penile spines presence assessment) or ultrasound examination or blood hormones analysis to define the reproductive status of a cat. However, although those methods are helpful and useful in most cases, sometimes they are not easy to be performed in non-sedated animals (i.e. vaginal cytology, penile spines presence assessment, less frequently also ultrasound examination). In addition, sometimes it is not easy for the clinicians to exactly define the reproductive status according to the degree of penile spines development or even on the basis of a single steroid hormone measurement, as reported also by Prochowska and Niżański (Journal of Feline Medicine and Surgery. 2022. 24, 837–846) who stated that: “Results of blood testosterone assessment can be difficult to interpret. While a high level (>1–9 ng/ml30) would be considered normal, a low baseline result (<1 ng/ml) does not exclude proper testicular function. Many mature, intact males show testosterone levels below the detectability threshold; this is due to pulsatile, episodic testosterone release”. In these cases, therefore, the long-term hormonal concentrations accumulated in coat and claws could help the clinician in the diagnosis, as a supplementary tool.

I disagree with the authors’ statement (Lines 66-71) that “Although sexual steroids were traditionally measured on blood, this matrix could be not considered as a first-line choice, because of invasiveness at collection and because of the only punctual information provided, not useful for monitoring hormones secreted in an episodic manner, like testosterone. Similarly, also other matrices like saliva, urines and feces, collectable without invasiveness, provide only punctual or short-term picture of measured hormones.” In many instances, we do need to measure steroid hormone levels from blood at a given point in time. While due to the episodic release of T or E2 this may not always be useful, even a baseline T could often differentiate intact from gonadectomized males, or a vaginal cytology can give accurate and useful information within minutes about the presence of estrogen influence in the body instead of blood E2 in

cats. Therefore, I question the rationale and practicality of measuring sex steroids in cats in these alternative matrices, and whether such long-term accumulated levels could have a diagnostic application.

AU: The rationale is based upon the characteristic pulsatile secretion of testosterone, that can fluctuate in the blood, so that a single measurement in blood may not provide useful information. The long-term accumulated hormones, could therefore provide a less fluctuating and more accurate information.

Also, we do not really know for how long and at what concentrations these matrices are able to store information about the long-term levels of these steroid hormones

AU: We strongly agree with the reviewer, and future studies are needed to better identify these aspects.

and how these levels are relevant to certain reproductive events and time points.

AU: This is the reason why we started from the unequivocally established sex and reproductive status of cats. This is the first step to set up possible reference values as from long time defined-values about the blood concentrations.

The authors’ results show that there are differences in hormone concentrations between the claw and dewclaw sites, but there are no differences between males and females in T levels in the claw/dewclaw matrices in contrast to the haircoat. This finding is in contrast to the authors’ conclusion that dewclaws are useful for E2 and T measurements (Lines 250-251), if it cannot even differentiate between males and females.

AU: The authors agree with the Reviewer: the sentence was misleading and it was rewritten.

Line 255-257: “In fact, the present study results showed that coat T concentrations measurement is suitable to distinguish between male and female pubertal cats during the breeding season.” Again, while it is scientifically possible, why should we use such matrices and measure hormones for this simple task?

AU: The sentence has been expanded to be clearer.

As for the selection of animals, the authors define their study population as “pubertal”, however, they have animals from 8-9 months to 10 years of age. By the authors’ description it seems that all animals were already postpubertal, as pubertal would rather refer to those animals that are going through puberty at that time. Also, the authors would need to specifically state whether these cats were reproductively normal, e.g. no cryptorchidism, DSD, ovarian/testicular pathology, etc, and whether or not they received (currently or in the past months) any medications /hormones that could interfere with reproductive status and hormone levels.

AU: Changes done according to the Reviewer suggestions.

Minor points of concern:
Lines 46-61 of Introduction: This description of physiology can be deleted. AU: Deleted as suggested
Line 64: HPG (gonadal) axis, not HPA.

AU: Sorry for the mistake. Corrected

Please revise the English, some sentences are difficult to understand.

AU: The whole text was revised as suggested

Line 165-166: Was the amount of coat/claw used for the extraction the same across samples? Please describe

AU: the information has been added to the manuscript

Table 1: Please clarify if concentrations are expressed on the freeze-dried weight.
AU: concentrations are expressed for mg of dry sample.
Table 2 and 3 could be deleted and the results denoted in Table 1 or explained within the text. AU: Done according to the Reviewer suggestions.

Round 2

Reviewer 3 Report

I would like to thank the authors for the revision and for their answers.

I still have a few comments and recommendations.

Line 50-53: “Although sexual steroids were traditionally measured in blood, this matrix could be not considered as a first-line choice, because of invasiveness at collection and because it provides only punctual information, not useful for monitoring hormones secreted in pulsatile manner, like testosterone.” I disagree with this sentence. The measurement of sexual steroids is still a very important and often first-line choice in many aspects of small animal reproduction. Referencing sexual steroids in general can also be misleading, e.g. progesterone is not known to fluctuate like testosterone; furthermore, stimulation tests can overcome the episodic release of T or E2. Please reconsider and reformulate this sentence.

Line 59: “long-term hormonal accumulation”. Include which hormones you are referencing from the cited studies.

Line 225-227: “The results, in fact, demonstrated once more, the usefulness of coat for studies monitoring function for extended period,..” – I think the results show that T and E2 can be measured in these matrices, but we do not know about the extent of accumulation. Please reformulate the sentence. The authors may also want to consider discussing briefly that, so far, it is not know for how long and at what concentrations these tissues are able to store information on these steroid hormones.

Line 266-267: … “because of the easiest collection, especially in awake animals, in comparison to the scrotal/abdominal area.” Delete this part of the sentence, it is a repetition of what was said above.

Line 263-265 and 268-269 about the interchangeability of dewclaws and coat for E2 and T within each sex. Although the authors did not find statistically significant differences in hormone concentrations between these matrices, the fact that dewclaws are not suitable to distinguish between sexes based on their T concentrations while coats appear to be suitable, the interchangeability of these matrices is not supported from a clinical or even scientific point of view. Please reformulate this part of the discussion, as in my opinion, it would be a more objective interpretation of your results.

Line 298: “Within each sex, E2 and T concentrations can be assessed on coat, but also on the dewclaws.” I recommend deleting this sentence.

Line 317-319, Acknowledgement: Please delete text/section or add acknowledgement.

Author Response

Authors are thankful to the Reviewer for his final considerations that improved the overall quality and legibility of the manuscript.

Please note that the computer used for the corrections runs a modified version of Word for Macintosh servers, thus the line numbering could be misaligned.

Below the answer to the comments:

Line 50-53: “Although sexual steroids were traditionally measured in blood, this matrix could be not considered as a first-line choice, because of invasiveness at collection and because it provides only punctual information, not useful for monitoring hormones secreted in pulsatile manner, like testosterone.” I disagree with this sentence. The measurement of sexual steroids is still a very important and often first-line choice in many aspects of small animal reproduction. Referencing sexual steroids in general can also be misleading, e.g. progesterone is not known to fluctuate like testosterone; furthermore, stimulation tests can overcome the episodic release of T or E2. Please reconsider and reformulate this sentence.

Sentence was totally reconsidered by the authors.

Line 59: “long-term hormonal accumulation”. Include which hormones you are referencing from the cited studies.

Added as requested.

Line 225-227: “The results, in fact, demonstrated once more, the usefulness of coat for studies monitoring function for extended period,..” – I think the results show that T and E2 can be measured in these matrices, but we do not know about the extent of accumulation. Please reformulate the sentence. The authors may also want to consider discussing briefly that, so far, it is not know for how long and at what concentrations these tissues are able to store information on these steroid hormones.

Sentence rephrased.

Line 266-267: … “because of the easiest collection, especially in awake animals, in comparison to the scrotal/abdominal area.” Delete this part of the sentence, it is a repetition of what was said above.

Deleted as requested.

Line 263-265 and 268-269 about the interchangeability of dewclaws and coat for E2 and T within each sex. Although the authors did not find statistically significant differences in hormone concentrations between these matrices, the fact that dewclaws are not suitable to distinguish between sexes based on their T concentrations while coats appear to be suitable, the interchangeability of these matrices is not supported from a clinical or even scientific point of view. Please reformulate this part of the discussion, as in my opinion, it would be a more objective interpretation of your results.

Rephrased as requested.

Line 298: “Within each sex, E2 and T concentrations can be assessed on coat, but also on the dewclaws.” I recommend deleting this sentence.

Authors agree about the fact that the two matrices are not interchangeable, however the cited hormones were really measured in the two matrices of both sexes in the present study.

Line 317-319, Acknowledgement: Please delete text/section or add acknowledgement.

Done.